# The Role of Inositol Hexakisphosphate Kinase in the Central Nervous System

**DOI:** 10.3390/biom13091317

**Published:** 2023-08-28

**Authors:** Tyler Heitmann, James C. Barrow

**Affiliations:** 1Department of Pharmacology and Molecular Sciences, School of Medicine, Johns Hopkins University, 725 North Wolfe Street Suite 300, Baltimore, MD 21205, USA; 2The Lieber Institute for Brain Development, 855 North Wolfe Street Suite 300, Baltimore, MD 21205, USA

**Keywords:** inositol pyrophosphates, synaptic vesicles, neuronal migration, drug discovery, CNS disease, psychiatry, neuroscience

## Abstract

Inositol is a unique biological small molecule that can be phosphorylated or even further pyrophosphorylated on each of its six hydroxyl groups. These numerous phosphorylation states of inositol along with the kinases and phosphatases that interconvert them comprise the inositol phosphate signaling pathway. Inositol hexakisphosphate kinases, or IP6Ks, convert the fully mono-phosphorylated inositol to the pyrophosphate 5-IP7 (also denoted IP7). There are three isoforms of IP6K: IP6K1, 2, and 3. Decades of work have established a central role for IP6Ks in cell signaling. Genetic and pharmacologic manipulation of IP6Ks in vivo and in vitro has shown their importance in metabolic disease, chronic kidney disease, insulin signaling, phosphate homeostasis, and numerous other cellular and physiologic processes. In addition to these peripheral processes, a growing body of literature has shown the role of IP6Ks in the central nervous system (CNS). IP6Ks have a key role in synaptic vesicle regulation, Akt/GSK3 signaling, neuronal migration, cell death, autophagy, nuclear translocation, and phosphate homeostasis. IP6Ks’ regulation of these cellular processes has functional implications in vivo in behavior and CNS anatomy.

## 1. Introduction

The inositol pyrophosphate pathway is made up of a number of polyphosphorylated inositol derivatives containing both mono and diphosphate groups which are interconverted by specific kinases and phosphatases (Figure 1) [1]. This pathway is of central importance for cell signaling in eukaryotic species. The enzyme that converts the fully monophosphorylated inositol, inositol hexakisphosphate (IP6), to the pyrophosphate, 5-diphospho-1,2,3,4,6-pentakisphosphate (5-IP7 or IP7), inositol hexakisphosphate kinase (IP6K) has three isoforms: IP6K1, IP6K2, and IP6K3. IP6K1 and IP6K2 are expressed ubiquitously throughout the body, whereas IP6K3 is expressed in higher levels in skeletal muscle, cardiac muscle, and the thyroid [2,3]. IP7 can be further phosphorylated to IP8 (1,5-bisdiphospho-2,3,4,6-tetrakisphosphate) and IP6 may also be phosphorylated on the 1-position to 1-IP7. Although the concentrations of 1-IP7 and IP8 are much lower than that of 5-IP7 in the cell, their role in cell signaling is significant and a burgeoning area of research [4].

IP6Ks transmit signals in the cell via three main mechanisms. Their reaction products, IP7 and IP8, can bind to target proteins altering their function, or they can transfer a phosphate group from their pyrophosphates to a pre-phosphorylated residue on a target protein resulting in a pyrophosphorylated residue. Additionally, the enzyme itself, IP6K, can bind to target proteins altering their function via protein–protein-interactions (PPI) [1].

Genetic and pharmacologic manipulation of IP6K activity affects a number of cellular processes including vesicular trafficking [5,6,7,8], insulin signaling [8,9,10,11,12], DNA damage and repair [13,14], and cell migration [13,15,16]. These cellular roles of IP6Ks have led to their study in the physiological processes of metabolism [10,12,17,18,19,20,21], blood coagulation [22], sperm production [11,23], phosphate homeostasis [22,24], fat accumulation [10,12,18,19], cancer [25], infection [26,27,28], bone loss [29], behavior [30,31], and tau phosphorylation [30]. The importance of IP6K in these processes has led to interest in the development of IP6K inhibitors with potential therapeutic value in diseases including diabetes, obesity, fatty liver disease, chronic kidney disease, osteoporosis, cancer, thrombosis, hyperphosphatemia, mood disorders, and Alzheimer’s disease [24,32,33,34,35].

The role of IP6Ks in metabolic disease has been well studied and reviewed in other articles [1]. For decades, the importance of IP6Ks in the brain has been slowly uncovered. Through mechanisms that regulate Akt/glycogen synthase kinase (GSK3) signaling, vesicular trafficking, cell migration, cell death, and nuclear translocation, IP6Ks have profound effects on behavior, motor function, and central nervous system (CNS) structure in vivo. In this review article, the growing body of literature is critcally analyzed and the importance of IP6Ks in the CNS is summarized.

## 2. Akt/Glycogen Synthase Kinase 3 (GSK3) Signaling

In the classical Akt/GSK3 signaling pathway, external stimulation of a G-protein coupled receptor (GPCR) or receptor tyrosine kinase (RTK) induces Akt association to the plasma membrane by binding phosphatidylinositol (3,4,5)-trisphosphate (PIP_3_). Phosphoinositide-dependent kinase-1 (PDK1) at the membrane then phosphorylates and activates Akt. Active Akt, along with a multitude of other functions, phosphorylates and inactivates the normally constitutively active GSK3 [36,37].

IP6Ks modulate this pathway in two distinct ways, both of which function to enhance GSK3 activity. First, IP7 competes for PIP_3_ binding to Akt, thereby preventing its association with the membrane and subsequent phosphorylation [10]. Secondly, IP6K1 binds directly to GSK3 preventing the inactivating phosphorylation by Akt and enhancing its activity (Figure 1) [30].

GSK3 is a well-established central signaling molecule, important in the normal function of neurons [38]. There is substantial evidence for GSK3 playing a role in psychiatric disease. GSK3 inhibition is postulated to play a role in the mechanism of action of antipsychotics, antidepressants, and lithium [37,39,40,41]. It was found that stimulation of D2 dopamine receptors results in Akt complexing with β-arrestin-2 and protein phosphatase 2 (PP2A), which deactivates Akt, therefore enhancing GSK3 activity [42]. Multiple studies have also found that increasing serotonergic tone through 5HT_1_ receptors results in the inhibitory phosphorylation of GSK3 [43,44,45,46,47,48].

Many psychiatric drugs operate though these mechanisms to regulate Akt/GSK3 activity. Antidepressants such as selective serotonin reuptake inhibitors (SSRIs) and monoamine oxidase inhibitors (MAOIs) that increase serotonergic tone inhibit GSK3 [44,45]. Amphetamine, which elevates dopaminergic tone, inhibits Akt and activates GSK3 [49,50,51]. Antipsychotics which block D2 receptors have the opposite effect, showing increased Akt activity and decreased GSK3 activity [44,52,53]. Finally, lithium has been shown to inhibit GSK3β directly in vitro, albeit at concentrations higher than those used clinically [54,55]. Perhaps more relevant is lithium’s ability to disrupt the Akt/β-arrestin-2/PP2A complex that regulates GSK3β activity [56]. Furthermore, IP6Ks have been shown to regulate synaptic vesicle recycling which has been proposed to be mediated in part through GSK3 signaling [6]. The mechanisms of Akt/GSK3 and their role in psychiatric illness have been thoroughly discussed in a number of notable review articles [38,57,58].

GSK3 has also been implicated in neurodegenerative disease. In Alzheimer’s disease, GSK3 has been shown to promote amyloid β production [59] as well as tau phosphorylation [60], as GSK3β is a major kinase that phosphorylates tau [61,62]. Both amyloid plaques and hyperphosphorylated tau neurofibrillary tangles are the main histopathological markers of Alzheimer’s disease. There is also evidence that GSK3 inhibition may be beneficial in multiple sclerosis, Parkinson’s disease, and other neurological diseases which have been summarized in other review articles [38,63,64].

Multiple campaigns by pharmaceutical companies have attempted to inhibit GSK3 for indications in the human CNS with little clinical success. Any potent chemical series that has good target exposure has been linked with toxicity as GSK3 is a central regulator in many processes [65,66,67,68,69]. A better therapeutic strategy may be to target upstream regulators of GSK3, such as IP6Ks to allow for a cell context-specific inhibition of GSK3. Targeting IP6Ks may have a larger therapeutic index as IP6K knockouts are relatively healthy compared to GSK3 knockouts which are embryonically lethal [70].

## 3. Synaptic Vesicle Regulation

The role of inositol pyrophosphates in the regulation of synaptic vesicle exo- and endocytosis has been studied since their discovery in the early 1990s. Since then, the work of multiple research groups has begun to elucidate the mechanisms of synaptic vesicle regulation by IP6Ks. IP6Ks also have a clear role in vesicular trafficking outside of the brain, notably in promoting insulin vesicle exocytosis [8,71]. The mechanisms of synaptic vesicle regulation work through multiple different pathways including the Rab3A GTPase, clathrin mediated endocytosis, synaptotagmin dependent exocytosis, and Akt/GSK3 signaling.

### 3.1. Synaptotagmin

Depolarization of the presynaptic neuron causes voltage gated calcium channels to open, resulting in an influx of calcium. On synaptic vesicles, the synaptotagmin C2A domain binds this calcium, which causes a conformational change that allows the C2B domain to bind phosphatidylinositol 4,5-bisphosphate (PIP_2_) in the plasma membrane as well as syntaxin, a plasma membrane protein. These interactions along with interactions between synaptobrevin on the vesicle and synaptosome-associated protein 25 (SNAP25) on the plasma membrane then allow for synaptic vesicle fusion with the plasma membrane (Figure 2) [72].

IP7 (and to a lesser extent IP6) inhibits exocytosis by binding to the synaptotagmin C2B domain polylysine sequence [5,73]. This interaction is not competitive with Ca^2+^ and has been shown to functionally inhibit neurotransmitter release in PC12 cells and primary cultured hippocampal neurons, likely via competing for PIP_2_ binding on the plasma membrane [5].

### 3.2. Rab3A/GRAB

Rab3A is a GTPase expressed specifically in synaptic vesicles of neurons that regulates vesicle exocytosis. GRAB (guanine exchange factor for Rab3A) binds and activates Rab3A via its coiled-coil domain, exchanging GDP for GTP [74]. Active, GTP-bound Rab3A is known to associate with synaptic vesicles, inhibiting their exocytosis. Upon GTP hydrolysis, Rab3A dissociates from the synaptic vesicle. Although the exact mechanisms of how Rab3A affects synaptic vesicle exocytosis are incompletely understood, it is known that Rab3A binds to the C2B domain of synaptotagmin on synaptic vesicles. This interaction inhibits the binding of synaptotagmin to syntaxin on the plasma membrane, and thereby exocytosis [75].

IP6K1 was found to bind to the coiled-coil domain on GRAB, competing with Rab3A, thereby keeping Rab3A in the inactive GDP-bound state, enhancing exocytosis. Functionally, in PC12 cells IP6K1 transfection results in decreased GTP loading of Rab3A and enhanced depolarization induced dopamine release [74]. Interestingly, the binding of IP6K1 to GRAB to enhance neuro-exocytosis seems to act counter to the action of its product IP7 binding to synaptotagmin to inhibit exocytosis. It is possible that these mechanisms act in different cell types, cellular contexts, or within different stages of synaptic vesicle exocytosis. Further work is needed to fully elucidate the role of IP6Ks in the exocytosis of synaptic vesicles.

### 3.3. Clathrin Adaptor Proteins (APs)

Clathrin-mediated endocytosis is the main route in which synaptic vesicles are reformed after exocytosis. Clathrin is a protein that forms a polyhedral lattice coat around a budding vesicle, providing it with structure during the process of endocytosis. Clathrin adaptor proteins (APs) mediate the association of the membrane with clathrin (Figure 2) [72]. APs associate with the membrane via interactions with phosphoinositide lipids such as PIP_2_ in the membrane as well as with transmembrane proteins [76]. Once a budding vesicle is coated with clathrin and APs, the final step of pinching off the vesicle is carried out by the motor protein dynamin [72]. Clathrin-mediated formation of synaptic vesicles can occur directly from the plasma membrane, or during more intense bouts of stimulation, it can occur from endosomes formed via activity dependent bulk endocytosis (ADBE). ADBE occurs after neurons fire rapidly, they endocytose a large endosome rather than small vesicles to rapidly internalize excess plasma membrane (Figure 2).

There are three subtypes of clathrin-associated APs: AP1, AP2, and AP3 [77]. AP2 facilitates synaptic vesicle endocytosis directly from the plasma membrane [78]. AP1 and AP3 facilitate vesicle formation indirectly via endosomes that are formed from ADBE [79]. AP2-mediated vesicle formation occurs more rapidly than AP3-mediated vesicle formation as it skips the large endosomal intermediate step. When AP3 is inhibited, endocytosis occurs, counterintuitively, faster. This is due to the redirection of endocytosis away from the slow AP3-mediated process and towards the fast AP2-mediated process [72,79,80].

In the early 1990s, it was initially discovered that IP6 binds to AP2 and AP3 [81,82]. These binding interactions function to inhibit clathrin coat assembly, inhibiting endocytosis. Subsequent studies revealed that IP7 binds to AP3 with a much higher affinity than IP6 (50 nM vs. 240 nM, respectively) [83]. It has been proposed that IP7 could be competing for PIP_2_ in the membrane binding to APs, inhibiting endocytosis [84]. Furthermore, upon binding AP3, IP7 pyrophosphorylates the beta subunit of AP3 [85]. This inhibitory pyrophosphorylation of AP3 by IP7 has been shown to have a functional role on Kif3A (kinesin motor proten family member 3A) in the release of HIV viral particles [85]. However, Kif3A motor proteins have important roles in dendritic architecture, cargo loading, and transport in neurons as well [86,87,88].

Lastly, it was recently discovered that IP6K1 binds to phosphoinositide 3-kinase (PI3K). Locally produced IP7 then binds to the RhoGAP domain of PI3K to disinhibit its interaction with the sodium/potassium ATPase. This recruits AP2 which promotes endocytosis of the pump and its downstream degradation. Deletion of IP6K1 causes a two-fold enrichment of the Na+/K+ ATPase in the plasma membrane. The Na^+^/K^+^ ATPase is not only critical in establishing electrochemical gradients in neurons, but also is expressed ubiquitously in eukaryotic cells [89].

In summary, IP7 binds and pyrophosphorylates AP3 to inhibit clathrin-mediated endocytosis. IP7 has also been shown, in specific situations, to promote AP2-mediated endocytosis indirectly via disruption of the Na^+^/K^+^ ATPase/PI3k interaction promoting the degradation of the ATPase.

### 3.4. Dynamin

IP6Ks, via a number of mechanisms, enhance dynamin-mediated endocytosis. IP6Ks enhance GSK3 activity directly and indirectly via Akt, as described above. This pathway is highly relevant in synaptic vesicle regulation as GSK3 is known to promote ADBE via phosphorylation of dynamin [90]. Furthermore, Akt was found to regulate ADBE via GSK3 during intense neuronal activity [91]. Interestingly, IP6 and IP7 were found to regulate dynamin as well via phosphorylation of pacsin/syndaptin I. This phosphorylation facilitates interaction between dynamin and pacin/syndaptin I, promoting assembly of the endocytic machinery [92].

### 3.5. Ex Vivo Mouse Experiments

The mechanisms by which IP6Ks participate in the regulation of synaptic vesicles are complex, as detailed above. IP7 inhibits exocytosis via synaptotagmin, and inhibits endocytosis directly via AP3 and indirectly via Akt/GSK3/dynamin. IP6K1 has additional protein interactions with GRAB that function to enhance exocytosis, and with GSK3 that function to inhibit endocytosis. Given this complex network of regulation, experiments aimed at measuring the effects of IP6Ks on synaptic vesicle dynamics must be carried out and interpreted with careful consideration.

Two such studies have been recently published by Park et al. in 2020 [7] and Li et al. in 2022 [6]. Both of these studies perform electrophysiology experiments in IP6K1 knockout (KO) and wild-type (WT) mouse brain slices and cultured neurons, looking at measures of exocytosis and endocytosis.

Park et al. found that during electrical stimulation of hippocampal slices from CRIPSR-generated IP6K1 KO mice, the rate of short-term facilitation and the paired plus ratio (PPR) was decreased in IP6K1 KO mice compared to WT. Additionally, they found that presynaptic release probability was increased in IP6K1 KOs. Interestingly, reintroduction of both WT and kinase inactive mutants of IP6K1 revealed that catalytic activity is required to rescue presynaptic release probability, but short-term facilitation was rescued by the kinase inactive mutant. Together these results suggest that IP6K1 and IP7 inhibit synaptic vesicle exocytosis in an activity-dependent manner [7].

Park et al. also found defects in synaptic vesicle endocytosis and recycling in IP6K1 KO hippocampal slices. Following synaptic vesicle exhaustion via excessive electrical stimulation, neurons were allowed to recover for 3 min via endocytosis and reformation of synaptic vesicles. At the end of the 3 min, WT neurons fully recovered their synaptic responses. IP6K1 KO synaptic responses recovered to only 60% of their initial amplitude, implying impaired synaptic vesicle resupply after intensive challenge. Furthermore, they applied pharmacological approaches to further characterize IP6Ks role in endocytosis. Folimycin inhibits the refilling of new synaptic vesicles with a neurotransmitter [93]. Dynasore is an inhibitor of dynamin, the motor protein responsible for pinching off new synaptic vesicles during endocytosis [94]. In these experiments, cells were stimulated at 20 Hz to establish a baseline, then treated with folimycin or dynasore for 10 min, then stimulated at 20 Hz again. The decrease in amplitude between baseline and after treatment represents the amount of endocytosis that occurred during the baseline stimulation. IP6K1 KO brain slices show no difference in amplitude between baseline and after treatment, implying impaired endocytosis during stimulation in IP6K1 KOs. This indicates that IP6K1 and IP7 are important in maintaining normal synaptic vesicle recycling [7].

Li et al. performed live cell electrophysiology imaging experiments using optical reporters of synaptic vesicles in cultured hippocampal neurons. They tagged the vesicular glutamate transporters VGLUT1 and VGLUT 2 with pHluorin optical reporter tags that are quenched in the low pH of synaptic vesicles but fluoresce when they are exocytosed. VGLUT1 is recycled into synaptic vesicles via the AP2 mediated mechanism of endocytosis directly from the plasma membrane. VGLUT2 is recycled via ADBE into endosomes then into synaptic vesicles via AP1- and AP3-mediated mechanisms. VGLUT2 can also be recycled via the AP2-mediated mechanism [6].

In these experiments, Li et al. showed that pharmacological inhibition of IP6Ks with a small molecule (TNP) [95], shRNA knockdown of IP6K1 or IP6K3, and genetic KO of IP6K1 or IP6K3 in cultured hippocampal neurons showed accelerated endocytosis of VGLUT2 but not VGLUT1. Similar results were seen in prolonged stimulation conditions, which showed an increase in the recycling pool in VGLUT2 but not VGLUT1. Consistent with these data, in brain slices, IP6K1 KO accelerates recovery of neurotransmitter release in VGLUT2 expressing synapses, but not VGLUT1 synapses. These results of enhanced endocytosis mirror those of AP3 inhibition, which redirects VGLUT2 to the faster AP2-mediated endocytotic pathway. Given that IP7 inhibits AP3 via pyrophosphorylation, these results are paradoxical; however, the other mechanisms by which IP6Ks regulate endocytosis such as GSK3/dynamin could be at play [6,90,91,92]. These results are in contrast to those in the Park et al. study which showed impaired endocytosis in IP6K1 KO hippocampal brain slices, albeit in different contexts [7]. Taken together these data suggest that IP6Ks regulate the molecular machinery involved with bulk endocytosis.

Li et al. also found in cultured hippocampal neurons that pharmacological inhibition of IP6Ks with TNP or KO of IP6K1 resulted in increased exocytosis of VGLUT1 and VGLUT2. Furthermore, in hippocampal brain slices of IP6K1 KO mice, there was an increase in action potential driven spontaneous excitatory post synaptic current (sEPSC) amplitude and frequency compared to WT [6]. Additionally, in contrast to Park et al. who found decreased synaptic facilitation and lower PPR in IP6K1 KOs, Li et al. found increased synaptic facilitation and a higher initial PPR in IP6K1 KO hippocampal brain slices [6,7]. Li et al.’s results of increased activity-dependent exocytosis may be explained in part by IP7′s inhibition of synaptotagmin [5,73]. It may be that in IP6K1 KOs, IP7 is decreased, synaptotagmin is disinhibited, and therefore exocytosis is enhanced.

Much work has been done to uncover the mechanisms by which IP6Ks regulate synaptic vesicle dynamics. Synaptotagmin, GRAB/Rab3A, clathrin AP3, GSK3, and dynamin are all known targets by which IP6Ks exert their control over synaptic vesicles. Furthermore, well thought out work has been done in brain slices and cultured neurons to characterize the functional significance of IP6Ks in synaptic transmission. This exciting area of research will continue to untangle the complex net of regulation that IP6Ks employ in the future.

## 4. Cell Migration

Focal adhesions are structures that connect the actin cytoskeleton with the extracellular matrix (ECM) and allow the cell to respond to extracellular chemical and mechanical stimuli [96]. Transmembrane integrins bind to talin which allows integrin to bind to the ECM, talin then binds a single F-actin filament [97]. Vinculin is then recruited to bind talin and additional F-actin filaments in a process known as maturation [98,99]. Maturation is advanced via phosphorylation of focal adhesion kinase (FAK) which promotes phosphorylation of α-actinin and paxillin, which then recruits vinculin to focal adhesions, among other roles. Activation of FAK also promotes focal adhesion turnover, which is critical for cytoskeletal remodeling during cell migration [99].

α-Actinin is a cytoskeletal protein whose major function is to crosslink other cytoskeletal proteins. It is known to crosslink actin filaments as well as vinculin and actin in focal adhesions. α-actinin is also known to interact with a number of other cytoskeletal proteins, membrane proteins, and ion channels [97].

The actin-related protein 2/3 (Arp2/3) is a protein complex responsible for the generation of actin networks at the leading edge of the cell which form lamellipodia protrusions, crucial in neuronal migration and synapse formation [100]. Coronin binds the p34 subunit of Arp2/3 and inhibits the formation of new actin filaments during remodeling [101,102].

Dynein is a microtubule based cytoskeletal motor protein complex important in retrograde transport of cargo. It also functions at the cell membrane where it tethers and generates pulling forces [103]. Dynein binds to dynactin via an interaction between dynein intermediate chain 2 (DIC2) and p150glued on dynactin. This interaction is required for dynein activity and recruits dynein to the leading edge of migrating cells [104].

In 2017, Fu et al. discovered that IP6K1 binds to α-actinin. Furthermore, they discovered that in IP6K1 KO and TNP treated cells, phosphorylation of α-actinin, FAK, and paxillin are markedly reduced. They then worked out that IP7 binds to FAK promoting its autophosphorylation, dimerization, and activation [102]. This allowed them to propose a model in which IP6K1 binds to α-actinin, coming into close proximity with FAK. IP7 is synthesized from IP6K1, which then binds FAK, inducing its autophosphorylation. pFAK can then go on to phosphorylate α-actinin and paxillin [15].

In 2023, the Fu group also showed that IP6K1 also binds to the actin-related protein 2/3 (Arp2/3) complex. IP7 synthesized by IP6K1 then binds to both the p34 subunit of Arp2/3 and coronin facilitating their interaction, presumably by acting as a “molecular glue”. This interaction inhibits the nucleation of new actin filaments by Arp2/3 at the leading edge of lamellipodia [102].

In 2019, they expanded these discoveries to include IP6K3′s role in focal adhesion dynamics. They discovered that IP6K3 binds to directly to DIC2 [16]. IP7 generated by IP6K3 then pyrophosphorylates serine 51 on DIC2 [28], promoting its interaction with p150glued on dynactin, recruiting the complex to the leading edge of the cell membrane [16].

Together these findings allowed the Fu and Snyder groups to propose the following compelling model for how IP6Ks regulate focal adhesion dynamics (Figure 3). IP6K1 binds α-actinin. Local IP7 binds FAK inducing its autophosphorylation and activation. pFAK then phosphorylates α-actinin and paxillin, promoting focal adhesion maturation and turnover. Some IP6K1 also binds to Arp2/3. Local IP7 then binds Arp2/3 and coronin, inhibiting new actin filament nucleation during remodeling [100,101,102]. Similarly, IP6K3 binds to DIC2. Local IP7 pyrophosphorylates DIC2, promoting interaction with dynactin, recruiting the complex to the leading edge of the membrane. At the membrane, dynein’s association with the focal adhesion complex along with its retrograde action on microtubules, may contribute to the migration of the cell. This model highlights IP6Ks role as a scaffolding protein and the sensitivity of local changes in IP6/IP7 concentration [15,16,28].

These mechanisms of IP6K’s regulation of focal adhesions and cell migration have functional in vivo implications in the brains and motor behavior of KO mice (described in detail below).

## 5. Cell Death

IP6K2 appears to control processes involved in autophagy [105], mitophagy [106,107], and apoptotic cell death [108,109]. These processes are also important in neurodegenerative CNS diseases such as Huntington’s disease (HD), amyotrophic lateral sclerosis (ALS), and Alzheimer’s disease. Some work, described below, has also been done to establish IP6K2′s role in neurodegeneration.

IP6K2 is typically expressed in the nucleus. In many mammalian cell lines, in response to many different stressors, IP6K2 is translocated from the nucleus to the cytoplasm, where it contributes to a rise in IP7 levels [108,109,110]. In HEK293 cells and ovarian carcinoma 3 (OVCAR3) cells treated with cell stressors staurosporine and cisplatin, respectively, green fluorescent protein (GFP)-tagged IP6K2 translocates from the nucleus to the cytoplasm and co-localizes with apoptotic mitochondrial stain Bax [108]. Another study found that IP6K2 promotes autophagosome formation, potentially through inhibition of mTOR signaling via IP7 inhibition of Akt [105].

Subsequent studies have linked the role of IP6K2 in autophagy and apoptosis to HD and ALS. Lymphoblast cells were taken from patients with HD and healthy controls. In HD lymphoblasts, IP6K2 was found mainly in the cytoplasm, compared to healthy controls where it was mainly found in the nucleus. As expected, this led to an increase in IP7 levels in HD lymphoblasts. Consequently, HD lymphoblasts showed a higher number of autophagosomes. Finally, HD lymphoblasts exhibited an increased level of cell death compared to healthy controls, which could be suppressed via siRNA knockdown of IP6K2 [109].

Nagata et al. theorize that in neurodegenerative disease, the accumulation of toxic proteins or external stressors results in the activation of IP6K2, which then promotes autophagosome formation to remove these toxic protein aggregates. Cell death may also be promoted by similar mechanisms if the toxic proteins or external stressors cannot be cleared by the cell [109].

In ALS and frontotemporal lobar degeneration with ubiquitin inclusions (FTLD-U), protein aggregates are found in the neuronal cytoplasm in the brain and spinal cord. The TAR DNA-binding protein 43 (TDP-43) is the major component of these protein aggregates and is thought to contribute to neuronal cell death in these patients [111,112]. In SHSY-5Y cells, transfected with both TDP-43 and IP6K2, IP6K2 translocates to the cytoplasm and colocalizes with TDP-43 aggregates, leading to an increase in cell death. In TDP-43 transfected cells, pAkt and casein kinase 2 (CK2) levels were increased. Double transfection with TDP-43 and IP6K2 reduced pAkt and CK2 levels to normal [110].

Increasing pAkt and CK2 may be a cellular defense mechanism in response to TDP-43 aggregates [110]. CK2 phosphorylates TDP-43 aggregates, leading to their degradation [113]. CK2 also phosphorylates IP6K2 leading to its degradation [114]. IP6K2 acts antagonistically to these mechanisms. In response to cell stressors, IP6K2 is activated via translocation to the cytoplasm, resulting in increased IP7, decreased pAkt and CK2, and cell death [110]. Thus, targeting IP6Ks, or even IP6K2 specifically, may be a therapeutically useful strategy for reducing cell death in neurodegeneration.

In contrast, some studies have shown that IP6K2 may actually be neuroprotective. In IP6K2 KO cerebella, there is increased mitochondrial fission and mitophagy along with reduced mitochondrial function due to defects in the electron transport chain and interaction with creatine kinase B (CK-B) [106,107]. These mitochondrial defects in IP6K2 KO cerebella may in part cause the neuronal cell death, and decreased synapse formation seen in these mice (described below).

## 6. Nuclear Translocation

Protein 4.1N is a neuron-enriched member of the protein 4.1 family with numerous known interacting partners, but generally its role is in protein scaffolding with membrane proteins. Functionally, 4.1N is important in synaptic plasticity, synaptic transmission, the cytoskeleton, cell proliferation, cell adhesion, and signal transduction [115]. The class of 4.1 proteins generally contains three conserved domains: the N-terminal 4.1-ezrin-radixin-moesin (FERM) domain, the internal spectrin–actin-binding (SAB) domain, and the C-terminal domain (CTD). In between these three conserved domains are the three non-conserved domains U1, U2, and U3 [115]. Furthermore, FERM domains are known to bind phosphoinositides such as PIP_2_ [116].

In response to nerve growth factor (NGF), 4.1N translocates to the nucleus where it mediates antiproliferative actions [117]. Nagpal et al. found that IP6K2 binds directly with 4.1N causing its translocation into the nucleus in response to NGF. In IP6K2 KOs, expression of several proteins was decreased such as Rab 5, SNX-27 (sorting nexin 27), synaptophysin, NGF, Fos-B, and Arc (activity-related cytoskeletal-associated protein), implying a role for the IP6K2/4.1N interaction in gene expression. As a result, IP6K2 KO mice show a number of defects in the cerebellum, described in detail below.

Additionally, Rao et al. discovered that IP6K2 interacts with liver kinase B1 (LKB1) and in an IP7-dependent manner causes phosphorylation of serine 428 on LKB1, which results in its nuclear translocation [118]. Although this study primarily focused on LKB1/IP6K2′s role in cancer metastasis, LKB1 is also known to play an important role in the CNS [119].

IP6K2 was found to be expressed selectively by enteric nervous system neurons. Furthermore, IP6K2 was found to regulate transcription of genes associated with mature neurons, stem cells, glial cells, and genes modulating neuronal differentiation and functioning in the enteric nervous system. Although this study only looked at most genes in the enteric nervous system, they showed that dopamine D5 receptor and the cholecystokinin B receptor were selectively upregulated in the enteric nervous system of IP6K2 KO mice and not in the CNS [3]. Further work needs to be done to investigate IP6K2’s role in gene expression in the CNS.

## 7. Phosphate Homeostasis

Phosphate is an essential component of many facets of cell signaling and physiology. The fundamental molecules of biology, DNA, RNA, ATP, phospholipids, and phosphoproteins, all employ phosphate groups for their function. Therefore, it is not surprising that the inositol pyrophosphates, which have the highest density of phosphates of any molecules found in nature, are key regulators of phosphate homeostasis.

In addition to the physiological role phosphate plays in bone health, platelet-mediated coagulation [22], and kidney health [24], phosphate dysregulation in the CNS can also lead to human disease. One such example is primary familial brain calcifications (PFBC), a neurodegenerative disease in which patients display bilateral basal ganglia calcifications that result in neuropsychiatric and movement dysfunction. Loss of function mutations in xenotropic and polytropic retrovirus transport 1 (*XPR1*) have been shown to cause PFBC, in addition to associations with other genes [120]. Interestingly, XPR1 is the only known cellular inorganic phosphate efflux transporter in humans [121].

IP6Ks have been shown to regulate XPR1 via the synthesis of IP7 and IP8. IP7/IP8 were shown to inhibit XPR1 via binding of its SPX domain [122,123]. The binding of inositol phosphates to SPX domains of numerous yeast and plant proteins to regulate inorganic phosphate is well studied. However, interestingly, XPR1 is the only known human protein with an SPX domain [121,123,124,125,126,127]. These early in vitro studies hint at a therapeutic strategy, where reduction in IP8 levels may restore XPR1 activity in patients with loss of function mutations in diseases of phosphate disregulation such as PFBC.

It is important to note that in the study by Li et al., techniques were used to specifically isolate the effects of IP7 versus IP8. Genetic manipulation or inhibition of IP6Ks will inevitably affect both IP7 and IP8 synthesis. The development of metabolically stable methylene bisphosphonate (PCP) analogs of IP7 and IP8, along with methods to deliver these chemical tools via liposomes, has allowed for the different effects of the inositol pyrophosphates to be studied [128,129].

## 8. Knockout Mice

IP6K1, 2, and 3 knockout mice show a number of behavioral phenotypes as well as structural brain abnormalities that can be explained by some of the biochemical roles of IP6Ks described above.

### 8.1. IP6K1

In addition to metabolic phenotypes such as reduced high fat diet-induced obesity and increased insulin sensitivity, IP6K1 KO mice also show CNS-relevant phenotypes.

In IP6K1 KO mice, during embryonic development and early post-natal life, impaired neuronal migration into cortical layers is seen. Furthermore, cysts are observed in cortical layers 2–5 from birth onwards. Interestingly, in this study, premature death was seen in IP6K1 KO mice [15]. Given IP6Ks and IP7′s role in regulating focal adhesions and cell migration described above, it is likely that disruptions in these same pathways are to blame for the structural impairments seen in IP6K1 KO mice [13,15].

Additionally, IP6K1 KO mice display altered behavioral phenotypes. When placed in a novel environment, KO mice show decreased locomotor activity compared to WT mice. In an accelerating rotarod test, KO and WT mice perform similarly, indicating that the decreased locomotion in a novel environment is not due to defects in strength or motor coordination. When dosed with D-amphetamine, both WT and KO mice show an increase in locomotion. However, the effect is significantly reduced in KO mice compared to WT from 30 to 90 min after treatment [30]. This phenotype is also seen in GSK3β haploinsufficient mice, indicating that IP6K1′s regulation of GSK3 could be contributing to this phenotype [49].

When tested in the Crawley Preference for Social Novelty Test (PSNT), IP6K1 KO mice show a reduced preference for interacting with an unknown mouse over a neutral object or a known mouse. In an open area social interaction test, an IP6K1 KO or a WT mouse is placed with an unknown C57BL/6 mouse. In this experiment, IP6K1 KO mice initiate significantly fewer social interactions with the C57BL/6 mouse than WT [30]. GSK3β haploinsufficient mice show enhanced sociability [40], whereas GSK3α KO mice show reduced social interactions [130]. Therefore, further work would need to be done to identify the mechanisms of IP6K1’s role in murine social behavior.

In another study, Kim et al. investigated the role of IP6K1 in memory. IP6K1 KO mice show lower pre-pulse inhibition while having no deficits in the Y-maze and elevated plus maze tests. These results indicate a role for IP6K1 in sensorimotor gating [31]. Reduced pre-pulse inhibition has also been seen in mice with mutant Rab3A [131], Syt1 [131], Akt1 [52], and GSK3β [41]. Therefore, the reduced pre-pulse inhibition seen in IP6K1 KO mice may be due to alterations in any one of, or all of, these pathways.

Finally, when re-exposed to a foot shock chamber that the mice had been shocked in 1 h prior, IP6K1 KO mice exhibited reduced freezing compared to WT, indicating impaired short-term memory in the context of fear. When re-exposed to the chamber 24h after the initial shock, there was no difference between WT and KOs. Furthermore, in acute brain slices, IP6K1 KOs showed no differences in long-term potentiation or long-term depression, indicating that only short-term memory is impaired [31]. Interestingly, dentate gyrus specific KOs of GSK3β also showed impaired fear memory [132].

### 8.2. IP6K2

IP6K2 KO mice have been shown to display cerebellar abnormalities. In the cerebellum, IP6K2 localizes almost exclusively to the granule cells along with its known interaction partner, protein 4.1N. In IP6K2 KO mice, the number of granule cells is reduced by 25% and the cerebellar molecular layer width is reduced by 30%. Additionally, EdU labeling of proliferating cells reveals defects in neuronal development with a 45% decrease in labeled cells in IP6K2 KO cerebella [133].

In Purkinje cells, deletion of IP6K2 resulted in decreased density of spines and cellular volume by 50–65%. Additionally, synapses between Purkinje cells and the parallel fibers of granule cells were decreased by 35–50% [133].

IP6K2 KO mice also displayed several motor abnormalities associated with cerebellar defects. In a rotarod test, KO mice displayed a significantly decreased latency to fall, indicating impaired motor coordination. In an open field environment, IP6K2 KO mice show significantly decreased locomotion. Gait analysis revealed decreased stride length and speed in the IP6K2 KO mice [133]. In total, IP6K2 deletion results in defects of cerebellar granule cell development, which leads to Purkinje cell abnormalities and motor dysfunction.

### 8.3. IP6K3

IP6K3 KO mice exhibit a number of structural and functional abnormalities of the brain. IP6K3 is highly expressed in the Purkinje cells and interneurons of the cerebellum as well as in the cortex, hippocampus, thalamus, and hypothalamus [134]. In IP6K3 KO mouse cerebella, the width of the molecular layer is decreased and the number of synapses is decreased by 50% [134]. GABAergic synapses as well as parallel and climbing fiber synapses were decreased in KO mouse cerebella [134]. Purkinje cells of KO mice display dendritic growth retardation, decreased cell size, as well as impaired migration into the Purkinje cell layer [16]. Other cerebellar cell types such as granule, Golgi, basket, and stellate cells appear normal [134]. In KO IP6K3 mouse embryos, EdU staining revealed retardation of neuronal migration [16]. Furthermore, much like IP6K1 KO mice [15], IP6K3 KO mice show impaired neuronal migration in cortical layers 2–5 [16].

IP6K3 KO mice accordingly display motor abnormalities associated with Purkinje cell dysfunction. In the rotarod test, KO mice show a decreased latency to fall, indicating impaired motor coordination. Gait analysis showed impaired overlap between hindpaws and forepaws. However, in an open field test, KO mice performed similarly to WT [134]. Given IP6K3, IP6K1, and IP7′s role in regulating dynein and focal adhesions (described above), these structural and functional abnormalities caused by impaired neuronal migration are likely due to IP6K’s role in regulating this pathway.

Finally, in humans, an exploratory study of single nucleotide polymorphisms (SNPs) of the IP6K3 gene revealed two SNPs in the promoter region that are associated with late onset Alzheimer’s disease. One of these SNPs showed an increase in promoter activity and is associated with a decreased disease risk for Alzheimer’s disease [135]. The other SNP did not affect promoter activity but was associated with an increased risk for Alzheimer’s disease. Future studies are required to fully understand IP6K’s role in human disease, but this study hints at a potential role for IP6Ks in neurodegeneration.

It is clear that deletion of IP6Ks has significant effects on mouse behavior and motor function. However, further investigation is required to directly link these phenotypes to IP6K’s regulation of Akt/GSK3, synaptic vesicle recycling, neuronal migration/focal adhesions, protein 4.1N, cell death/apoptosis, phosphate homeostasis, or a yet to be discovered biochemical role of IP6Ks.

## 9. Conclusions and Future Directions

As described above, there is a growing body of literature establishing IP6Ks role in the central nervous system. Decades of molecular biology, genetics, neuroscience, and mouse behavioral work have provided a foundational understanding of IP6K’s role in the brain. Genetic manipulation in vitro and in vivo has begun to demystify the isoform specific roles IP6K1, 2, and 3.

It is clear that all three isoforms of IP6Ks, along with their products IP7 and IP8 play an important role in the central nervous system. Synaptic vesicle exo- and endocytosis, Akt/GSK3 regulation, neuronal migration, cell death, and nuclear translocation have all proven to be key cellular processes regulated by IP6Ks. These cellular processes have been shown to have in vivo implications in mouse behavior, motor function, and gross brain structure. While this work has obvious implications for human health and disease, there are still many unanswered questions surrounding IP6K’s role in the brain. The use of kinase inactive mutants and genetic knockouts have provided useful tools for deciphering the roles of IP6Ks versus those of IP7/IP8. Nevertheless, the continued development of novel chemical tools and genetic models is the clear path forward to begin answering some of these unanswered questions.

Pharmacologic targeting of IP6Ks in the CNS is still in its infancy. To date, the only known inhibitors of IP6Ks do not cross the blood–brain barrier (BBB). Furthermore, the most potent of these inhibitors are not isoform selective [24,32,33,35,136]. Currently, no structural information is known about the three mammalian IP6K isoforms aside from the AlphaFold predicted structures. The AlphaFold predicted structures for IP6Ks appear to be relatively trustworthy for the well-ordered nucleotide binding site; however, the alogrithm is unable to predict the 3D structure for the disordered G-loop. Having a high-resolution structure of IP6Ks would no doubt aid in chemical probe discovery efforts.

Any efforts to develop chemical tools should consider side effects and potential off-targets. Inositol kinases share many structural features with protein kinases despite the difference in their targets of phosphorylation [137]. Although non-trivial, previously developed peripherally restricted IP6K inhibitors have demonstrated the feasibility of attaining selectivity over non-inositol kinases [24,32,33,35,138]. Some of these compounds even further showed varying degrees of selectivity within IP6K isoforms [32,35,138]. Tools that differentiate the roles of IP7 and IP8 are also needed to more fully determine the individual roles of these two molecules [139].

Any campaign to target the kinases and phosphatases in the inositol phosphate pathway should carefully consider the effects of inhibition on levels of all of the inositol phosphates. Inhibition of IP6Ks would result in decreases in both IP7 and IP8 concentrations; therefore, the biological effects of both IP7 and IP8 should be conisdered during compound development. BBB-penetrating, isoform selective IP6K inhibitors would allow for the investigation of the acute effects of IP6K inhibition in the brain, in addition to holding therapeutic potential for a number of CNS diseases.

An IP6K PROTAC (Proteolysis Targeting Chimera) degrader would be a useful chemical tool, alongside a classical enzymatic inhibitor of IP6Ks, to tease out the importance of protein scaffolding roles versus IP7/IP8-related roles in observed phenotypes. A classical inhibitor would only prevent the synthesis of IP7/IP8, whereas a PROTAC degrader would abolish both scaffolding interactions and IP7/IP8 synthesis. While it is unlikely, but not impossible, for a PROTAC degrader to cross the BBB [140], this chemical tool would be useful in studying IP6Ks in vitro in a wide array of different cell lines without the need for genetic manipulation.

In this review article, a number of processes that IP6Ks affect have been outlined. Some of these processes such as synaptic vesicle dynamics, Akt/GSK3 regulation, and cell death seem to present IP6Ks as a promising target for treating CNS disease. However, other processes that IP6Ks regulate, such as neuronal migration and development, may be deleterious to alter in vivo. It is important to note, however, that pharmacological inhibition of IP6Ks in adults would occur long after brain development and neuronal migration is complete. It is unclear to what degree the behavioral phenotypes seen in IP6K KO mice are caused by defects in neuronal migration versus the other processes such as synaptic vesicle regulation, Akt/GSK3 modulation, and cell death. The development of a BBB-penetrating IP6K inhibitor could provide answers to this question and shine light on the mechanisms of IP6K’s role in behavior and psychiatric disease.

More nuanced genetic modifications of IP6Ks will also provide key insights into IP6K’s role, not only in the brain, but in other somatic tissues as well. IP6K kinase-inactive, mutant mice could allow us to decipher the roles of IP6K scaffolding interactions versus its enzymatic synthesis of IP7 and IP8 [141]. Brain-specific KOs of the IP6Ks would provide evidence that the behavioral changes seen in KO mice are specific to changes in the brain. Inducible KOs of IP6Ks would allow for mouse brains to fully develop before IP6K KO, in order to study IP6K disruption in the absence of defects in neuronal migration. The sustained research by teams across the globe will continue to elucidate the complex roles of IP6Ks in the central nervous system.

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
