# Peer review of "The Role of Inositol Hexakisphosphate Kinase in the Central Nervous System"

_biomolecules, 2023, doi:10.3390/biom13091317_

Round 1

Reviewer 1 Report

The authors have provided a largely appropriate review of the literature with regards to capturing what is known about the roles of IP6Ks in the CNS. So a number of my suggestions are relatively minor. However, there are a couple of major points which should definitely be addressed before this work can be acceptable for publication. In particular, they should discuss the role of IP6Ks in IP8 synthesis in relation to inorganic phosphate homeostasis.  

Major Points

1.
The authors are dismissive of the importance of IP6Ks in the pathway of synthesis of IP8, a molecule with eukaryote-wide functionality in regulating cellular inorganic phosphate homeostasis: PMID 37189392, 32019887, and 10.7554/elife.87956.1.

In particular: lines 38 and 39. “IP7 can be further phosphorylated to IP8. . . however, the concentrations of IP8 are much lower than that of 5-IP7 in the cell.” This is neglectful of the biological significance of IP8 based solely on its relatively low cellular levels. By way of comparison, the concentration of IP8 is equivalent to that of the calcium-mobilizing agonist IP3, which no-one has ever dismissed as being insufficiently abundant to be important to cell biology. The same respect should be given to IP8 too.

Furthermore, as pointed out by Fiedler and colleagues (PMID: 37189392), “In many cases it is difficult to attribute an observed phenotype to [IP7 or IP8], because genetic deletion of IP6Ks inevitably reduces the levels of both.” Indeed, Saiardi and colleagues (PMID: 31186349) mistakenly claimed IP7 regulates cellular phosphate homeostasis in a study in which IP6Ks were knocked out, before later work reported that IP8 is the active messenger.

Given the published importance to the CNS of cellular phosphate influx and efflux  transporters (PMID: 26923164, 25938945, PMID: 35063775 etc) shown elsewhere to be regulated by IP8, this topic deserves its own section in this review.
Additionally, by virtue of IP8 acting independently of IP7, inhibitors of IP6Ks have the potential for off-target effects; this point needs mentioning.

At the same time, the authors should correct other instances where the significance of IP6Ks in IP8 synthesis is neglected:

Line 531: “IP6Ks, along with their product IP7 play an important role in the central nervous system”.
Lines 537 to 539: “kinase inactive mutants and gene knockouts have provided useful tools for deciphering the roles of IP6Ks versus that of IP7” (why not IP8 too?)
Line 553: “ an [IP6K] inhibitor would only prevent the synthesis of IP7” (wrong! It would also inhibit IP8 synthesis).

2.
The final section headed ‘conclusions and future directions’ is somewhat too brief.
In this forward-looking section, there could be a little more discussion of the viability of developing an IP6K inhibitor with low off-target effects against not just other members of the IPK family, but also the entire protein kinase family; these two groups of proteins share many key structural features (PMID: 30392847) and so might also be targets of IP6K inhibitors; this point should be noted. There are data in the authors’ cited references that speak to the goal of attaining such specificity.    

Highly relevant to the subject of off-target effects of IP6K inhibitors is their impact on IP8 synthesis; IP8 and IP7 each have separate biological roles (see major point 1 above). This section is a singularly appropriate home for this discussion.  

Minor points

1.    Lines 160-166. Could the authors offer some analysis/rationalization of the (counterintuitive?) opposing effects of IP6K1 and IP7 on exocytosis?

2.    Lines 67-118. In this section on GSK3, it would be useful to cite work described by ref 5 in this section concerning the impact of IP6Ks on synaptic vesicle recycling, which is hypothesized to involve regulation of GSK-3 (as the authors themselves mention in the sections that follow).

3.    Lines 365-367 and lines 394-395. The authors seem to imply that stress-dependent translocation of IP6K2 from the nucleus to the cytoplasm contributes to a rise in ip7 levels. It would be useful to explain how Nagata and colleagues determined those two events are directly connected. Why is nuclear activity of IP6K2 not able to enhance cellular ip7 levels?

4.    Lines 514-519. A slightly more detailed description of the data in citation 124 is warranted, and perhaps the value of this study is somewhat over-stated by omission of these details. First, one of the two SNPs was actually associated with a decreased susceptibility to late onset Alzheimer’s disease, which seems worth some mention. As for the other SNP – the nature of the promoter also deserves to be noted. Finally, the authors of 124 are careful to admit this was an ‘exploratory’ study with a relatively small cohort and limited statistical analysis. Thus, the authors of this review should at least acknowledge ref 124 is preliminary in nature. As for the claim that study 124 “points at a physiologically relevant role for IP6Ks in neurodegeneration.” Perhaps the more cautious “hints at a potential role for IP6Ks. . . ” would be more appropriate.  

5.    lines 545. The authors should add ‘mammalian’ prior to “IP6K isoforms”, since the crystal structure of an amoeboid IP6K has been solved. There could also be a little more detail concerning why the AlphaFold modeling is of limited use: the algorithm is unable to reliably predict the 3D structures of the disordered G-loop, which in other kinases optimize the alignment of the triphosphate of ATP for substrate phosphorylation. The AlphaFold predictions of the rest of the well-ordered nucleotide binding site can be expected to be rather more trustworthy for molecular modeling.

Reviewer 2 Report

More nuanced genetic modifications of IP6Ks will also provide key insights into IP6Ks role in the brain. In this manuscript, the authors sumarized the role of IP6Ks in Central Nervous System, which would contribute to functional implications in vivo in behavior and CNS anatomy. But I have several following concerns:

1. Abbreviations should be defined when they first appear in the text.

2. Please graphically show sequence or structural differences among isoforms of IP6K.

3. In the main text, the reference order should be placed before the punctuation mark..

4. Please provide the publication licenses for Figures 2 and 3.

5. Since Inositol Hexakisphosphate Kinase inhibitor is a kinase, whether there are some modulators abou this kinase and how about their biological effects against diseases generated in Central Nervous System.

6. The author's title is "The Role of Inositol Hexakisphosphate Kinase in the Central Nervous System", a suggested way to write it is to introduce the role of the gene in physiology and pathology separately, rather than the pathway, protein, and cellular events in the present paper. It gives the impression that the logic is a bit muddled.

7. Please unify the format of references in the article, including the author's name, the case of words in the title of the article, the writing of the name of the journal, and the page number.  

Minor editing of English language required.

Round 2

Reviewer 1 Report

The authors have commendably addressed all editorial concerns

Reviewer 2 Report

The authors have addressed all my comments. I recommend accepting it in current status.